# The intensity of male-male competition may affect chemical scent constituents in the dark ventral patch of male Iberian red deer

Eva de la Peña[1]*, José Martín[2], Juan Carranza[1]

**1** Wildlife Research Unit (UIRCP), Universidad de Córdoba, Córdoba, Spain, **2** Departamento de Ecología Evolutiva, Museo Nacional de Ciencias Naturales, CSIC, Madrid, Spain

* evadelapenha@gmail.com

**Data Availability Statement:** There are not ethical or legal restrictions on sharing the used data set in this work, we have deposited ours in FIgshare

## Abstract

During the mating season, Iberian red deer males (*Cervus elaphus hispanicus*) present a large visible dark ventral area in their abdomen. This characteristic dark-haired area is formed by the impregnation of the hair with sprayed urine and gland secretions and contains volatile compounds that can be used in intraspecific communication. Here, we used gas chromatography-mass spectrometry (GC-MS) to describe the lipophilic chemicals from the dark ventral patch of males from different populations with different levels of intrasexual competition. Amongst all the compounds found, *m*-cresol, benzoic acid, cholesterol and 4-hydroxy-benzenopropanoic acid were the most abundant. The proportions of these compounds varied with age as well as with the level of intra-sexual competition, independently of age. In particular, red deer males experiencing higher intra-sexual competition had lower proportions of aromatic compounds (especially *m*-cresol) but higher proportions of carboxylic acids on their dark bellies. Males in a high male-male competition situation, invest in volatile compounds that can reveal their age, dominance status and condition, and that, in addition, enhance this signal. On the contrary, males from low intra-sexual competition populations have chemical profiles more characteristic of young individuals. This research shows a first glance of how secretion of volatile compounds of male deer can be modulated due to the intensity of male-male competition in the population.

## Introduction

Communication is considered as the process by which the sender uses signals or carries out displays that are specially intended to modify the behaviour of the receivers [1]. Signals evolve through a kind of selection that maximizes the effectiveness in changing the receiver's behaviour [1–2]. For this, and given the limitations of different environments, animals should base their communication in different sensory channels [3].

Chemical signals play an important role in intraspecific communication of many animals [4–5]. The usefulness of chemical signals lies on their tenacity; once the signal is transmitted, the information remains in the environment even though the emitter has gone away. Chemical

**Funding:** Financial support came from projects CGL2013-48122-P and CGL2016-77052-P to JC. The funders had no role in study design, data collection and analysis, decision to publish, or preparation of the manuscript.

**Competing interests:** The authors have declared that no competing interests exist.

signals operate even under bad light conditions and can be received over long distances [5–6]. Chemical signals are also honest signals due to their costs; only good quality individuals can deal with the cost of their production with some specific characteristics [7].

Chemical communication in vertebrates is considered one of the most important systems due to its ability to transmit very specific information [4–5]. Many mammals have diverse glands that produce chemical signals that are usually incorporated in scent marks with the aim of signalling to conspecifics territoriality or, dominance, or to attract partners [4,8]. For example, in Eurasian deer species, interindividual differences in the secretions from different glands (e.g. preorbital, metatarsal and interdigital) are related to age, dominance status and sexual traits [9]. In white-tailed deer (*Odocoileus virginianus*), secretions from the interdigital glands and from the forehead region are relevant in sexual communication and individual recognition. Those glands produce secretions full of volatile compounds that deer spread over the hair rubbing their legs [10–11]. Urine plays also a key role in deer intraspecific communication. The presence and concentration of some urinary compounds can vary depending on season, sex, reproductive status and social rank in males of white-tailed deer [12–13]. Deer often urinate over their tarsal glands during the mating season, which may help them to scent-mark territories and to wear some chemical scent on their bodies that may signal social status [13–14]. It is considered that tarsal gland scent production is originated from the interactions between urinary compounds, microbial decomposition and gland secretions [13–14].

The Iberian red deer (*Cervus elaphus hispanicus*) is a polygynous species, and adult males usually defend harems or mating territories on the preferred females' locations [15–17]. The rutting behaviour of male red deer includes an extensive repertoire of communicative behaviours directed either to rival males or to females. The most widely known signal is the highly evident roaring acoustic call [18–19], with roaring rate per unit of time being related to body condition [20]. In addition, there are also highly evident visual signals such as the size of the whole body and antlers, displayed in lateral view to rivals during the parallel walk preceding antler fighting [21]. These signals are intended to avoid the direct confrontation with other males in order to not suffer damages.

However, chemical communication is poorly investigated in the red deer, although it is supposed to also play a relevant role during the rut. Different types of volatile compounds, such as short chain carboxylic acids and aromatic compounds are incorporated to the urine of deer [22]. During the mating season, the secretion of those compounds increases in adult males, whilst levels remain constant in females [22]. During the rut, red deer males show a large and conspicuous dark ventral patch of hair. This patch is not a permanent trait, as its size increases during the rutting season, reaching up to 70 cm long in mature males (Carranza *et al.*, unpublished data). The dark coloration is caused by an exogenous agent; male red deer spray their bellies with urine and the elevated concentrations of catecholamines and derived catechols, found in the urine, oxidize in contact with the air and produce the black coloration in the hair [23]. The deposition of chemical compounds on the hair of the belly might increase the volatilizing surface, improving the detectability of the chemical signal, as reported in other mammals [24]. In fact, the maximal surface occupied by this dark ventral patch increases with age, until deer reach the age of six years, when the dark patch remains stable in size, following the same pattern than other morphological traits related to intrasexual competition between males, such as antler size or body weight (Carranza *et al.*, unpublished data). The dark ventral patch hair of the Iberian red deer males is impregnated with a wide diversity of chemical compounds that may have originated from several sources, such as the urine or the sebaceous glands of the skin [25]. In particular, heterocyclic aromatic organic compounds, steroids, carboxylic acids and alcohols are found, among others, in the belly black spot, but not in other hair areas. Some of these are strongly odorous compounds, which proportions are also related to male age and,

thus, they may have a role in chemical communication during the rutting season signalling age or dominance [25]. However, it is still unknown whether these compounds that impregnate the belly black spot may have additional signalling functions.

In Southern Spain, red deer populations occur in hunting estates under two different management regimes. Some estates are fenced by 2 m high stock-proof wire mesh, while other estate areas are unfenced allowing free movements of deer between estates [26–28]. Fenced hunting estates reduce hunting pressure on young males, allowing them to reach maturity. In contrast, in unfenced hunting estates few stags reach old age. Thus, in unfenced estates, hunting pressure is not only on old deer with the best trophy antlers, but on almost every male above two years of age, except yearlings, as it is illegal to shoot them [28]. As a consequence of such contrasting management, the red deer populations within unfenced areas have mostly young males and strongly female-biased sex ratios, compared to the situation in fenced areas where there are more old than young males and less females [26,28]. Therefore, in fenced estates, males experienced a higher level of intra-sexual competition for access to females than in unfenced estates, where competition is low as virtually all males can mate even if they are subadult [26–27,29]. These different population scenarios offer the opportunity to test the effects of the level of male-male competition on the investment on sexual chemical signals.

Previous studies in red deer have shown that male intra-sexual competition level in a population promotes increased sex-trait expression, both antlers length and the dark ventral patch size (Carranza, unpublished data). Under a high male-male competition situation, there is a positive and close relationship between levels of testosterone andcortisol, which show the costs in terms of stress arising from investment in secondary sexual traits and reproductive effort (De la Peña, unpublished data). Social environment also mediates the link between mating effort and parasite burden, being less parasitized the males from low competition populations (De la Peña, unpublished data).

Thus, we hypothesized that differences in the social environmental context due to the different intensities of male-male competition may also result in differences in chemical signalling. To test this, we used gas chromatography-mass spectrometry (GC-MS) to describe the lipophilic fraction of the chemical compounds found in the dark ventral patch of male deer from different populations with different levels of male-male competition during the rutting season. Our aim was to examine whether the variation in the presence and relative abundance of volatile compounds in the dark ventral patch of males may be related to the age and the intensity of intrasexual competition in each population.

## Material and methods

### Study area

Extremadura Government granted permission to our research group to collect samples during hunting actions in public and private estates. The study was conducted in Mediterranean ecosystems in Southwestern Spain, in several hunting estates from 'Sierra de San Pedro' and 'Sierra de Monfragüe' (Cáceres province, W. Spain). The predominant vegetation in the study areas includes, in the highest elevation areas, Mediterranean scrub (*Cistus spp.*, *Erica spp.*, *Genista hirsuta*, *Lavandula spp.*), as well as forests with several tree species (*Arbutus unedo*, *Quercus spp.*, *Olea europaea*, *Phyllirea spp.*). While at the lower elevation areas, there are open pastures with scattered holm oak trees (*Quercus spp.*), known as "dehesas". The thicket and the forest are often used as shelter by the deer, while in the open "dehesas" deer are grouped during the breeding season.

From an ecological point of view, both populations are considered two metapopulations. Thus, each population is allocally settled forming an open population, fragmented and

discontinued, that occupies different patches with independent dynamics, but not related to the other metapopulation (unpublished data). Their members have a common evolutionary history, and there is genetic exchange. The building of human infrastructures, such as roads and fences, that become geographic barriers, and the management for hunting activities, resulted in that each of these regions was subdivided into different populations or hunting estates that range in size between 750 and 3000 Ha and are under two types of management regimes (fenced vs. unfenced; see introduction) [28]. From now on, we will refer to fenced and unfenced estates as populations with high (HC) and low (LC) levels of male-male competition, respectively.

## Hair samples collection

We took hair samples of male red deer from individuals harvested during the hunting activities performed in these estates between 15th October 2005 and 6th November 2005 to minimize time spent since the rutting season started and sampling collection. In the study area, the rutting season ussually takes place at the end of September. Thus, samples were obtained at most one month after the mating season started. We sampled a total of 84 Iberian red deer males from three estates of Sierra de San Pedro (two fenced and one unfenced estate) and two estates of Sierra de Monfragüe (one fenced and one unfenced estate). We collected hair samples from the dark patch of the belly, using a pair of clean scissors. Samples were directly transferred and preserved inside clean glass vials, closed with a Teflon septum. The same procedure was used for control vials, in which hair was not introduced, in order to know the contaminants that could appear during the handling of analytical procedures of because of the environment. Vials were kept cold in a portable cooler until they were taken to the laboratory where they were frozen at -20˚C until being analysed a few weeks after collecting hair.

To determine the age of the individuals, we collected their jaws, and, in the laboratory, we counted the dental cement layers of the interradicular pad of the first molars. This is a common procedure used by different authors in red deer and other species [30–31]. Average age (± SE) of individuals from which hair samples were taken was 3.0 ± 0.3 years (range = 2–11 years).

## Chemical analyses

We transferred a small amount of each hair sample to a clean glass vial. We added 250 μl of *n*-hexane (Sigma, capillary GC grade) and closed each vial with a teflon septum. The solution was homogenized using a vortex for 1 min. The samples were then kept in the freezer at -20˚C for 24 h. After that time, we extracted the supernatant with a glass Pasteur pipette and transferred it to a clean vial closed with the same teflon septum. The samples were analysed with a gas chromatograph (GC) (Finnigan-ThermoQuest Trace 2000), equipped with a poly (5% diphenyl / 95% dimethylsiloxane) column (Supelco, Equity-5, 30 m length x 0.25 mm ID, 0.25 μm film thickness), and coupled to a mass spectrometer (MS) (Finnigan-ThermoQuest Trace 2000) as a detector. We injected 2 μl of each sample in splitless mode with an inlet temperature of 250˚C. The GC temperature program was as follow: initial temperature of 45˚C maintained for 10 min, and then increased 5˚C/min until reaching 280˚C, which was maintained for 15 min. The carrier gas was helium, with a flow speed of 30 cm/s. Electron impact ionization (70 eV) was carried out at 250˚C. Mass spectra fragments below m/z = 39 were not collected. Impurities that were identified in the control vials processed and analysed with the same method were not considered. We made a first tentative identification of the compounds by comparing the mass spectra of the samples with those of the computerized NIST/EPA/NIH 2010 mass spectral library. Whenever it was possible, we confirmed initial identifications by

comparison of their spectra and retention times with those of authentic standards (from Sigma-Aldrich Chemical Co) coinjected under the same conditions.

## Statistical analyses

We used the software Xcalibur 1.2 (Finnigan Corp.) to monitor and manage the data generated by the GC-MS equipment. To determine the relative amount of each compound, we measured the total ion current (TIC) by automatically integrating the peak area of each compound in the chromatogram. It is not possible to establish direct comparisons between compounds using the absolute quantities of each one in the same sample since, each compound contains a different amount of each current ion separately depending on its polarity. However, it is possible to compare the mixture of compounds between different samples and the relative proportions of a compound, or a major class of compounds, in a sample, since the same error may be present in all the calculations [32]. To correct the problem of non-independence of proportions [33], we used the compositional analysis, consisting in logit transforming the proportion data by taking the natural logarithm of proportion/(1 – proportion).

The software PRIMER V6.1.13 [34] and PERMANOVA + V1.0.3 [35] were used to test for differences between the chemical profiles. We calculated the Euclidean distances between each pair of individual samples to build-up a matrix of similarities that formed the basis of our data. We used permutational multivariate variance analyses (PERMANOVA) [36] based on the Euclidean similarity matrix, using 999 permutations, to analyse whether chemical profiles varied between age categories and between levels of male-male competition. Pairwise post-hoc comparisons were made with permutation tests. Differences between ages and intensities of competition were further investigated using canonical analyses of principal coordinates (CAP) [37].

In addition, we carried out a principal component analysis (PCA) with all the compounds with relative proportions greater than 0.5% to reduce the number of variables, losing the least possible amount of information. The principal components (PC) scores defined by the PCA were used as new variables in general linear models (GLMs) to test for differences in proportion of compounds between age categories and between levels of male-male competition. For all statistical analyses a significant level of 5% was established and tests were carried out with the STATISTICA 8.0 software [38]. Dataset used in this study is available in the following link: https://doi.org/10.6084/m9.figshare.8985869.v1.

## Results

We found a total of 67 compounds in the dark ventral patch of hair of 84 sampled Iberian red deer males (Table 1). Aromatic heterocyclic organic compounds with benzene rings were predominant (53.90%), followed by steroids (22.72%), alcohols (7.82%), carboxylic acids (6.80%) and their ethyl esters (5.09%), ketones (2.34%) and minor components such as terpenoids (0.77%), sulfones (0.11%), ethers (0.01%) and others. On average, the four most abundant compounds found were *m*-cresol (23.40%), cholesterol (19.58%), benzoic acid (5.22%), and 4-hydroxy-benzenepropanoic acid (4.70%).

The PERMANOVA based on the matrix of similarities that compared the relative proportion of compounds found in deer's hair showed that there were significant differences between the chemical profiles of individuals directly related to their age (Pseudo $F_{2,85} = 3.39$, $P = 0.001$) (see Table 1). These differences were explained because the chemical profiles of the youngest individuals (ages between 2 and 3 years old) differed significantly from those of older deer, aged between 4 to 6 ($t = 1.88$, $P = 0.002$) and 7 years and above ($t = 2.05$, $P = 0.001$). However, these two older age classes did not significantly differ between them ($t = 1.18$, $P = 0.12$). A

**Table 1. Relative proportions (mean ± SE) of compounds found in hexane extracts of hair from the dark ventral patch of Iberian red deer males (*Cervus elaphus hispanicus*) depending on different age categories and different scenarios of male-male competition.**

| RT (min) | Compound | Low competition 2–3 years n = 30 | Low competition 4–6 years n = 6 | High competition 2–3 years n = 25 | High competition 4–6 years n = 12 | High competition ≥ 7 years n = 11 | All samples n = 84 |
|---|---|---|---|---|---|---|---|
| 7.8 | 4-Hydroxy-4-methyl-2-pentanone (= diacetone alcohol) | 0.18 ± 0.13 | - | 0.47 ± 0.38 | - | 0.71 ± 0.71 | 0.30 ± 0.15 |
| 8.3 | Hexanoic acid | 0.02 ± 0.02 | - | - | - | - | 0.01 ± 0.01 |
| 10.5 | Butyl ether | 0.03 ± 0.02 | - | - | - | 0.01 ± 0.01 | 0.01 ± 0.01 |
| 12.0 | 2,5-Dimethylpyrazine | 0.01 ± 0.01 | - | - | - | 0.01 ± 0.01 | 0.01 ± 0.01 |
| 12.2 | Dimethyl sulfone | 0.23 ± 0.12 | 0.35 ± 0.35 | - | - | 0.01 ± 0.01 | 0.11 ± 0.05 |
| 12.9 | 2(5H)-Furanone | - | - | - | - | 0.01 ± 0.01 | 0.01 ± 0.01 |
| 14.7 | Benzaldehyde | 0.05 ± 0.04 | - | - | - | 0.01 ± 0.01 | 0.02 ± 0.01 |
| 16.0 | Phenol | 0.10 ± 0.09 | 0.18 ± 0.18 | 0.04 ± 0.04 | 0.15 ± 0.15 | 2.18 ± 1.60 | 0.37 ± 0.22 |
| 16.7 | Trimethylpyrazine | 0.04 ± 0.04 | - | 0.18 ± 0.18 | - | 0.01 ± 0.01 | 0.07 ± 0.06 |
| 17.3 | 1,2,3-Benzothiadiazole | 0.02 ± 0.02 | - | - | - | 0.26 ± 0.19 | 0.04 ± 0.03 |
| 18.4 | Benzeneacetaldehyde | - | - | - | - | 0.09 ± 0.08 | 0.01 ± 0.01 |
| 19.4 | 2-Methylphenol (= *o*-cresol) | - | - | - | - | 0.10 ± 0.10 | 0.01 ± 0.01 |
| 19.7 | 3-Methylphenol (= *m*-cresol) | 41.84 ± 4.52 | 14.09 ± 2.97 | 22.83 ± 4.00 | 0.95 ± 0.66 | 3.96 ± 2.70 | 23.40 ± 2.66 |
| 20.2 | 2-Methoxyphenol | 0.05 ± 0.03 | 1.43 ± 0.86 | 0.25 ± 0.17 | 0.50 ± 0.36 | 0.04 ± 0.04 | 0.27 ± 0.10 |
| 21.0 | Phenylethyl alcohol | 1.40 ± 0.74 | 0.67 ± 0.67 | 0.11 ± 0.08 | - | 3.22 ± 1.86 | 1.00 ± 0.37 |
| 21.3 | Cyclohexanecarboxylic acid | 1.90 ± 0.64 | 2.37 ± 0.90 | 0.45 ± 0.30 | 2.41 ± 1.25 | - | 1.32 ± 0.32 |
| 21.7 | Cyclohexanecarboxylic acid, ethyl ester | 0.87 ± 0.38 | 3.89 ± 2.42 | 0.37 ± 0.23 | - | 1.36 ± 0.92 | 0.88 ± 0.27 |
| 22.9 | 4-Ethylphenol | 2.04 ± 0.78 | 1.16 ± 1.16 | 4.82 ± 1.33 | 7.41 ± 5.43 | 1.28 ± 0.72 | 3.47 ± 0.92 |
| 22.9 | Benzoic acid, ethyl ester | 0.37 ± 0.21 | 0.20 ± 0.20 | 0.19 ± 0.13 | 0.63 ± 0.63 | 0.58 ± 0.58 | 0.37 ± 0.14 |
| 23.0 | Nonanol | 0.28 ± 0.19 | 0.34 ± 0.22 | 0.94 ± 0.69 | 3.54 ± 3.14 | - | 0.91 ± 0.50 |
| 23.6 | 2-(2-Butoxyethoxy) ethanol | 0.51 ± 0.36 | 0.40 ± 0.26 | 0.96 ± 0.61 | - | 5.70 ± 5.21 | 1.25 ± 0.72 |
| 23.9 | Benzoic acid | 3.54 ± 1.41 | 5.26 ± 3.65 | 2.44 ± 1.69 | 4.43 ± 4.43 | 16.99 ± 9.67 | 5.22 ± 1.63 |
| 25.6 | 4-(2-Propenyl)-phenol | 0.20 ± 0.11 | 0.12 ± 0.12 | 0.22 ± 0.22 | - | - | 0.15 ± 0.08 |
| 25.7 | 4-Propylphenol | 2.50 ± 0.80 | 3.12 ± 1.68 | 0.90 ± 0.37 | 0.23 ± 0.23 | 1.67 ± 1.18 | 1.64 ± 0.37 |
| 26.0 | 4-Hydroxycyclohexanone | 2.26 ± 0.65 | 2.58 ± 0.94 | 1.10 ± 0.40 | 1.08 ± 0.53 | 1.32 ± 0.79 | 1.64 ± 0.30 |
| 27.1 | 1-(2-Hydroxy-5-methylphenyl) ethanone (= acetyl-*p*-cresol) | 3.49 ± 1.43 | 0.53 ± 0.36 | 2.47 ± 0.87 | 1.03 ± 0.63 | 3.37 ± 2.30 | 2.61 ± 0.65 |
| 28.1 | Benzenepropanoic acid | 0.43 ± 0.22 | - | 1.75 ± 0.86 | 0.17 ± 0.17 | 1.17 ± 0.90 | 0.85 ± 0.30 |
| 28.2 | Benzenepropanoic acid, ethyl ester | 0.19 ± 0.13 | - | 1.29 ± 1.21 | 0.04 ± 0.04 | - | 0.46 ± 0.36 |
| 29.1 | 2-Methyl-1,3-benzenediol | - | - | 1.08 ± 0.43 | 1.54 ± 1.04 | 6.70 ± 3.71 | 1.42 ± 0.56 |
| 29.4 | 4-Methoxy-1,3-benzenediamine | - | - | 0.01 ± 0.01 | - | 0.45 ± 0.43 | 0.06 ± 0.06 |
| 31.1 | 2-(3-Methyl-1,3-butadienyl)-1,3,3-trimethylcyclohexanol | 1.36 ± 0.38 | 4.19 ± 2.85 | 2.67 ± 0.86 | 2.22 ± 0.95 | 3.98 ± 2.15 | 2.42 ± 0.47 |
| 31.5 | Tetradecanol | 0.62 ± 0.28 | 2.04 ± 1.22 | 1.14 ± 0.66 | 2.84 ± 1.50 | 4.93 ± 3.99 | 1.76 ± 0.61 |
| 32.2 | Tridecanoic acid | 0.64 ± 0.47 | 0.00 ± | 4.83 ± 1.77 | 9.39 ± 4.29 | 3.53 ± 1.99 | 3.47 ± 0.90 |
| 32.3 | Hexadecanol | 0.57 ± 0.34 | 1.83 ± 1.19 | 2.33 ± 0.99 | 1.92 ± 1.18 | 0.88 ± 0.88 | 1.42 ± 0.39 |
| 33.4 | 4-Pentylphenol | 0.09 ± 0.09 | - | - | - | - | 0.03 ± 0.03 |
| 33.5 | 4-(4-Hydroxyphenyl)-2-butanone | 0.31 ± 0.22 | - | 0.11 ± 0.08 | 0.55 ± 0.55 | 0.22 ± 0.22 | 0.25 ± 0.12 |
| 33.7 | 3-Methylquinoline | 0.16 ± 0.16 | 0.56 ± 0.56 | 0.25 ± 0.15 | - | 0.07 ± 0.07 | 0.18 ± 0.08 |
| 34.3 | Benzoquinone | 0.06 ± 0.06 | 1.32 ± 1.03 | - | - | - | 0.12 ± 0.08 |
| 34.5 | Methylnaphthalene | 1.08 ± 0.55 | 2.00 ± 1.18 | 0.36 ± 0.25 | 0.34 ± 0.34 | 0.07 ± 0.07 | 0.69 ± 0.23 |
| 35.6 | 4-Hydroxybenzenepropanoic acid | 4.29 ± 1.12 | 10.43 ± 5.28 | 2.88 ± 0.79 | 6.14 ± 2.37 | 5.22 ± 4.43 | 4.70 ± 0.89 |
| 36.7 | 2-Ethylcyclohexanone | 0.32 ± 0.20 | 0.93 ± 0.60 | 1.30 ± 0.63 | 0.80 ± 0.69 | 0.03 ± 0.03 | 0.69 ± 0.23 |
| 38.1 | 17 hydroxy-17-metyl-3-androsta-1,4-dien | 0.26 ± 0.14 | - | 0.16 ± 0.11 | - | 0.06 ± 0.06 | 0.15 ± 0.06 |

(*Continued*)

**Table 1.** (*Continued*)

| RT (min) | Compound | Low competition | | | | | | High competition | | | | | | | | | All samples | | |
|---|---|---|---|---|---|---|---|---|---|---|---|---|---|---|---|---|---|---|---|
| | | 2–3 years n = 30 | | | 4–6 years n = 6 | | | 2–3 years n = 25 | | | 4–6 years n = 12 | | | ≥ 7 years n = 11 | | | n = 84 | | |
| 38.8 | Tetradecanoic acid, ethyl ester | 0.25 | ± | 0.14 | 0.27 | ± | 0.27 | 0.11 | ± | 0.09 | | - | | | - | | 0.14 | ± | 0.06 |
| 40.1 | Tetradecanoic acid | 0.40 | ± | 0.37 | | - | | 0.21 | ± | 0.14 | | - | | 0.50 | ± | 0.50 | 0.27 | ± | 0.15 |
| 41.5 | 9-Hexadecenoic acid | 0.07 | ± | 0.07 | | - | | 0.10 | ± | 0.08 | | - | | 0.08 | ± | 0.07 | 0.06 | ± | 0.03 |
| 42.0 | Pentadecanoic acid, ethyl ester | 0.08 | ± | 0.08 | | - | | | - | | | - | | | - | | 0.03 | ± | 0.03 |
| 42.1 | Hexadecanoic acid | 0.06 | ± | 0.06 | | - | | 0.57 | ± | 0.43 | 0.22 | ± | 0.22 | 1.74 | ± | 0.77 | 0.45 | ± | 0.17 |
| 42.4 | Hexadecenoic acid, ethyl ester | 0.18 | ± | 0.15 | | - | | 0.09 | ± | 0.09 | | - | | | - | | 0.09 | ± | 0.06 |
| 42.8 | Hexadecanoic acid, ethyl ester | 1.43 | ± | 0.40 | 0.53 | ± | 0.53 | 2.32 | ± | 0.75 | 3.51 | ± | 1.79 | 3.45 | ± | 1.71 | 2.19 | ± | 0.43 |
| 45.5 | 9-Octadecenoic acid | 0.07 | ± | 0.07 | | ± | | 3.31 | ± | 1.24 | 3.64 | ± | 1.50 | 3.37 | ± | 1.43 | 1.97 | ± | 0.49 |
| 45.8 | 9,12-Octadecadienoic acid, ethyl ester | 0.08 | ± | 0.05 | 0.22 | ± | 0.22 | 0.67 | ± | 0.49 | 0.64 | ± | 0.64 | 0.88 | ± | 0.61 | 0.45 | ± | 0.19 |
| 45.9 | Octadecanoic acid | 0.17 | ± | 0.12 | | - | | 1.49 | ± | 1.49 | | - | | 0.14 | ± | 0.10 | 0.52 | ± | 0.44 |
| 46.0 | 9-Octadecenoic acid, ethyl ester | 1.36 | ± | 0.48 | 0.95 | ± | 0.95 | | - | | | - | | | - | | 0.55 | ± | 0.19 |
| 46.4 | Octadecanoic acid, ethyl ester | 1.23 | ± | 0.33 | 0.65 | ± | 0.65 | 0.63 | ± | 0.26 | 0.17 | ± | 0.17 | 0.60 | ± | 0.44 | 0.78 | ± | 0.16 |
| 47.1 | Eicosanoic acid, ethyl ester | 0.67 | ± | 0.36 | 0.44 | ± | 0.44 | | - | | | - | | 0.25 | ± | 0.25 | 0.30 | ± | 0.14 |
| 47.8 | Octadecanol | 0.03 | ± | 0.03 | | - | | 1.97 | ± | 0.77 | 1.88 | ± | 0.87 | 0.96 | ± | 0.96 | 0.99 | ± | 0.30 |
| 49.8 | Docosanoic acid, ethyl ester | 0.09 | ± | 0.07 | 1.94 | ± | 1.76 | 0.87 | ± | 0.33 | 0.79 | ± | 0.52 | 0.45 | ± | 0.30 | 0.60 | ± | 0.18 |
| 51.1 | Eicosanol | 0.18 | ± | 0.10 | | - | | 0.45 | ± | 0.31 | 0.49 | ± | 0.49 | 0.50 | ± | 0.50 | 0.33 | ± | 0.14 |
| 52.2 | Androstane-3,17-dione | 2.04 | ± | 0.53 | 0.99 | ± | 0.99 | 0.42 | ± | 0.32 | 0.76 | ± | 0.67 | | - | | 1.03 | ± | 0.25 |
| 52.4 | 3-Hydroxyandrostan-17-one | 0.02 | ± | 0.02 | | - | | 1.64 | ± | 0.81 | 0.97 | ± | 0.88 | 0.90 | ± | 0.90 | 0.75 | ± | 0.30 |
| 54.0 | Docosanol | 0.11 | ± | 0.07 | | - | | | - | | | - | | 0.85 | ± | 0.85 | 0.15 | ± | 0.11 |
| 56.3 | Squalene | 0.24 | ± | 0.17 | 2.30 | ± | 2.11 | 1.41 | ± | 0.67 | 0.32 | ± | 0.28 | 0.40 | ± | 0.40 | 0.77 | ± | 0.27 |
| 57.4 | Cholesta-4,6-dien-3-ol | 0.61 | ± | 0.23 | | - | | 2.93 | ± | 1.11 | 1.26 | ± | 0.64 | 0.57 | ± | 0.40 | 1.34 | ± | 0.37 |
| 58.1 | 4-Octyl-N-(4-octylphenyl) benzenamine | 0.47 | ± | 0.38 | | - | | 0.42 | ± | 0.42 | 0.18 | ± | 0.18 | 0.01 | ± | 0.01 | 0.32 | ± | 0.19 |
| 60.4 | 17-Hydroxy-1,17-dimethylandrostan-3-one | 0.53 | ± | 0.25 | 1.01 | ± | 1.01 | 1.09 | ± | 0.46 | 0.18 | ± | 0.18 | | - | | 0.61 | ± | 0.18 |
| 61.2 | Cholesterol | 19.21 | ± | 3.68 | 18.91 | ± | 4.74 | 23.38 | ± | 4.02 | 22.26 | ± | 6.07 | 9.43 | ± | 4.94 | 19.58 | ± | 2.12 |
| 64.2 | Cholest-5-en-3-one | 0.75 | ± | 0.46 | 1.68 | ± | 1.68 | 3.21 | ± | 1.12 | 0.26 | ± | 0.26 | 0.82 | ± | 0.70 | 1.49 | ± | 0.42 |

CAP analysis classified 72.7% of the chemical profiles into their correct age categories using leave-one-outcross-validation and $m = 19$ axes ($\delta_1^2 = 0.63$, $P = 0.001$).

Considering individuals from all age categories, there were also significant differences in chemical profiles between the two different intensities of male-male competition (*Pseudo* $F_{1,86} = 5.87$, $P = 0.001$) (see Table 1), with 86.4% of the chemical profiles being correctly classified into their correct category of intensity of competition by a CAP analysis with $m = 18$ axes ($\delta_1^2 = 0.65$, $P = 0.001$).

However, to jointly analyse the effect of intrasexual competition level and age, and since in the LC situation there were no individuals of 7 or more years of age, we further considered only those samples corresponding to the ages 2–3 and 4–6 for both levels of male-male competition. Significant differences in chemical profiles were found between LC and HC populations (*Pseudo* $F_{1,72} = 3.91$, $P = 0.001$) and between age categories (*Pseudo* $F_{1,72} = 2.60$, $P = 0.004$), being the differences between levels of male-male competition independent of age (competition x age interaction, *Pseudo* $F_{1,72} = 0.93$, $P = 0.52$). A new CAP analysis classified 84.2% of the chemical profiles into their correct category of intensity of competition with $m = 18$ axes ($\delta_1^2 = 0.65$, $P = 0.001$).

Furthermore, comparing the relative proportions of the major classes of compounds between age classes (restricted to the ages 2–3 and 4–6) and categories of male-male

competition, we found that the proportion of aromatic compounds decreased in older males (GLM, age: $F_{1,72} = 7.26$, $P < 0.01$) but it was significantly higher in LC males than in HC males (66.8 ± 3.9% vs. 45.0 ± 3.7%; competition: $F_{1,72} = 9.80$, $P = 0.002$) independently of their age (age x competition: $F_{1,72} = 0.08$, $P = 0.78$). Conversely, the proportion of fatty acids did not vary significantly with age (GLM, age: $F_{1,72} = 0.01$, $P = 0.92$) but it was significantly lower in LC males than in HC males (1.2 ± 0.7% vs. 10.7 ± 2.0%; competition: $F_{1,72} = 18.55$, $P < 0.0001$) independently of their age (age x competition: $F_{1,72} = 0.74$, $P = 0.39$). However, no significant differences related to age or the level of male-male competition were found in the proportions of steroids (age: $F_{1,72} = 0.25$, $P = 0.62$; competition: $F_{1,72} = 0.31$, $P = 0.58$; age x competition: $F_{1,72} = 0.11$, $P = 0.74$), alcohols (age: $F_{1,72} = 0.65$, $P = 0.42$; competition: $F_{1,72} = 3.24$, $P = 0.076$; age x competition: $F_{1,72} = 1.87$, $P = 0.18$), ethyl esters of fatty acids (age: $F_{1,72} = 0.09$, $P = 0.76$; competition: $F_{1,72} = 0.19$, $P = 0.66$; age x competition: $F_{1,72} = 0.03$, $P = 0.86$) or ketones (age: $F_{1,72} = 1.56$, $P = 0.21$; competition: $F_{1,72} = 0.19$, $P = 0.66$; age x competition: $F_{1,72} = 0.04$, $P = 0.85$).

Furthermore, we compared with one-way ANOVAs the relative proportions of the four most abundant compounds impregnating hair of the dark ventral patch of deer between male-male competition levels (restricted to the two age categories found in both populations: 2–3 and 4–6 years). The proportion of *m*-cresol decreased in older males (GLM, age: $F_{1,72} = 21.18$, $P < 0.0001$) but it was significantly higher in LC males than in HC males (competition: $F_{1,72} = 15.42$, $P = 0.0002$) independently of their age (age x competition: $F_{1,72} = 1.62$, $P = 0.21$) (Table 1). However, no significant differences related to age or the level of male-male competition were found in the proportions of cholesterol (age: $F_{1,72} = 0.04$, $P = 0.85$; competition: $F_{1,72} = 0.04$, $P = 0.85$; age x competition: $F_{1,72} = 0.01$, $P = 0.93$), 4-hydroxy-benzenepropanoic acid (age: $F_{1,72} = 1.44$, $P = 0.23$; competition: $F_{1,72} = 0.31$, $P = 0.58$; age x competition: $F_{1,72} = 0.21$, $P = 0.64$) or benzoic acid (age: $F_{1,72} = 0.31$, $P = 0.58$; competition: $F_{1,72} = 2.40$, $P = 0.12$; age x competition: $F_{1,72} = 0.03$, $P = 0.85$) (Table 1).

The PCA on relative proportion of compounds yielded seven PCs with eigenvalues greater than 1, which together accounted for 49.48% of the variance (Table 2). We compared the relative proportions of the compounds described by the different PCs among the different age categories. In this case, the oldest individuals were also included, so three age categories were considered (2–3, 4–6 and equal or older than 7). Significant differences among age classes were found in PC3 (GLM, $F_{2,85} = 3.55$, $P = 0.03$). This indicated that the levels of 2-(2-butoxyethoxy) ethanol and 2-methyl-1,3-benzenediol were lower in young individuals (2–3 years), increasing in older males (Fig 1C). There were also significant differences among age categories in PC7 (GLM, $F_{2,85} = 14.20$, $P < 0.0001$), indicating that proportions of phenylethyl alcohol and hexadecanoic acid were higher in males of 7 years old and above (Fig 1G). There were no significant differences among age classes in the proportions of the other compounds defined by the remaining PCs (GLMs, $P > 0.87$ in all cases).

Comparing the PC scores between the two intensity levels of male-male competition (LC vs. HC), there were significant differences in the relative proportions of the chemical compounds in the hair defined by PC1 (GLM, $F_{1,86} = 10.39$, $P = 0.002$), PC3 ($F_{1,86} = 6.56$, $P = 0.01$), PC6 ($F_{1,86} = 14.34$, $P = 0.0001$), and PC7 ($F_{1,86} = 6.02$, $P = 0.02$) (Fig 1), respectively. Thus, the relative proportions of tridecanoic acid, hexanoic acid ethyl ester, octadecenoic acid, octadecanol, 3-hydroxy-5-androstan-17-one (PC1) (Fig 1A), cyclo-hexanecarboxylic acid ethyl ester, nonanol, hexadecanol (PC3) (Fig 1C), phenyl-ethyl alcohol and hexadecanoic acid (PC7) (Fig 1G) in the hair of the dark ventral patch were significantly higher in HC males than in LC males. Conversely, 9-octadecenoic acid ethyl ester and androstane-3,17-dione (PC6) (Fig 1F) appeared in significantly lower proportions in HC males than in LC males. No significant

**Table 2. Principal components analysis for compounds found in hexane extracts of hair from the dark ventral patch of Iberian red deer males (*Cervus elaphus hispanicus*).** Only compounds with relative proportions greater than 0.5% were used. Correlations between variables (compounds) and the principal components greater than 0.50 are marked in bold.

| | PC1 | PC2 | PC3 | PC 4 | PC5 | PC6 | PC 7 |
|---|---|---|---|---|---|---|---|
| 3-Methylphenol (= *m*-cresol) | 0.45 | 0.13 | 0.37 | 0.19 | 0.11 | -0.27 | 0.03 |
| Phenylethyl alcohol | 0.09 | 0.00 | 0.13 | -0.09 | 0.06 | -0.08 | **0.60** |
| Cyclohexanecarboxylic acid | 0.08 | -0.05 | -0.41 | 0.07 | 0.24 | -0.16 | -0.30 |
| Cyclohexanecarboxylic acid, ethyl ester | 0.10 | -0.15 | **-0.51** | -0.04 | 0.09 | -0.14 | 0.08 |
| 4-Ethylphenol | 0.08 | 0.17 | 0.06 | 0.16 | 0.09 | 0.21 | 0.05 |
| Nonanol | 0.09 | 0.05 | **-0.64** | 0.02 | 0.25 | 0.07 | -0.13 |
| 2-(2-Butoxyethoxy) ethanol | 0.11 | 0.06 | 0.04 | **-0.90** | 0.07 | 0.04 | -0.04 |
| Benzoic Acid | 0.16 | 0.07 | 0.11 | -0.10 | -0.27 | -0.16 | -0.18 |
| 4-Propylphenol | 0.14 | **-0.83** | 0.14 | 0.06 | 0.08 | 0.07 | -0.11 |
| 4- Hydroxycyclohexanone | 0.14 | **-0.82** | -0.04 | 0.04 | 0.03 | -0.11 | 0.06 |
| 1-(2-Hydroxy-5-methylphenyl) ethanone (= acetyl-*p*-cresol) | 0.01 | **-0.80** | -0.01 | 0.07 | -0.09 | 0.14 | 0.08 |
| Benzenepropanoic acid | -0.08 | -0.02 | 0.10 | 0.02 | 0.28 | 0.12 | 0.37 |
| 2-Methyl-1,3-benzenediol | -0.11 | 0.06 | 0.02 | **-0.93** | -0.00 | 0.05 | 0.08 |
| 2-(3-Methyl-1,3-butadienyl)-1,3,3-trimethylcyclohexanol | 0.14 | -0.48 | -0.22 | 0.09 | 0.01 | 0.38 | 0.35 |
| Tetradecanol | 0.03 | 0.20 | -0.19 | 0.02 | 0.02 | 0.01 | -0.19 |
| Tridecanoic acid | **-0.75** | 0.11 | 0.08 | -0.03 | 0.01 | 0.12 | -0.17 |
| Hexadecanol | 0.16 | 0.09 | **-0.79** | 0.08 | -0.18 | 0.06 | 0.14 |
| Methylnaphthalene | 0.15 | -0.20 | -0.11 | 0.03 | -0.02 | 0.26 | -0.28 |
| 4-Hydroxybenzenepropanoic acid | 0.16 | -0.25 | 0.04 | -0.01 | 0.08 | 0.47 | -0.26 |
| 2-Ethylcyclohexanone | -0.21 | -0.01 | -0.34 | 0.07 | -0.12 | 0.02 | -0.11 |
| Hexadecanoic acid | 0.05 | -0.11 | -0.16 | -0.10 | -0.21 | 0.04 | **0.69** |
| Hexadecanoic acid, ethyl ester | **-0.83** | 0.08 | 0.07 | 0.03 | 0.06 | -0.29 | 0.12 |
| 9-Octadecenoic acid | **-0.52** | 0.07 | 0.00 | 0.05 | -0.37 | 0.07 | 0.25 |
| 9-Octadecenoic acid, ethyl ester | 0.04 | -0.02 | 0.00 | 0.08 | 0.13 | **-0.75** | -0.08 |
| Octadecanol | **-0.91** | 0.08 | 0.07 | 0.00 | 0.05 | 0.03 | -0.03 |
| Docosanoic acid, ethyl ester | -0.39 | 0.13 | 0.07 | 0.04 | -0.16 | 0.15 | -0.05 |
| Androstane-3,17-dione | 0.03 | 0.18 | 0.14 | 0.14 | 0.03 | **-0.63** | -0.07 |
| 3-Hydroxyandrostan-17-one | **-0.82** | 0.05 | 0.03 | -0.04 | 0.06 | 0.07 | -0.07 |
| Squalene | 0.09 | 0.08 | -0.04 | 0.06 | **-0.78** | 0.11 | 0.06 |
| Cholesta-4,6-dien-3-ol | 0.13 | 0.17 | -0.16 | 0.21 | 0.37 | 0.31 | 0.40 |
| 17-Hydroxy-1,17-dimethylandrostan-3-one | 0.08 | 0.01 | -0.30 | 0.10 | -0.03 | -0.47 | 0.15 |
| Cholesterol | 0.02 | 0.31 | 0.04 | 0.21 | 0.38 | 0.23 | 0.03 |
| Cholest-5-en-3-one | 0.10 | -0.02 | -0.00 | 0.13 | **-0.65** | 0.04 | 0.04 |
| Eigenvalues | 4.10 | 2.70 | 2.11 | 2.05 | 1.94 | 1.74 | 1.70 |
| % Variance explained | 12.42 | 8.18 | 6.38 | 6.20 | 5.88 | 5.28 | 5.14 |

differences were found in the proportions of the remaining compounds defined by the other PCs (GLMs, *P* > 0.42 in all cases) (Fig 1B, 1D and 1E).

## Discussion

We found many chemical compounds of different classes in the hair of the ventral dark patch of Iberian red deer males, some aromatic compounds being the most abundant ones, such as *m*-cresol and benzoic acid, and steroids such as cholesterol. Previously, *m*-cresol and benzoic acid have already been showed as the principal volatile compounds present in the secretion from the tail gland of male red deer [39]. We found that the proportion of some of these

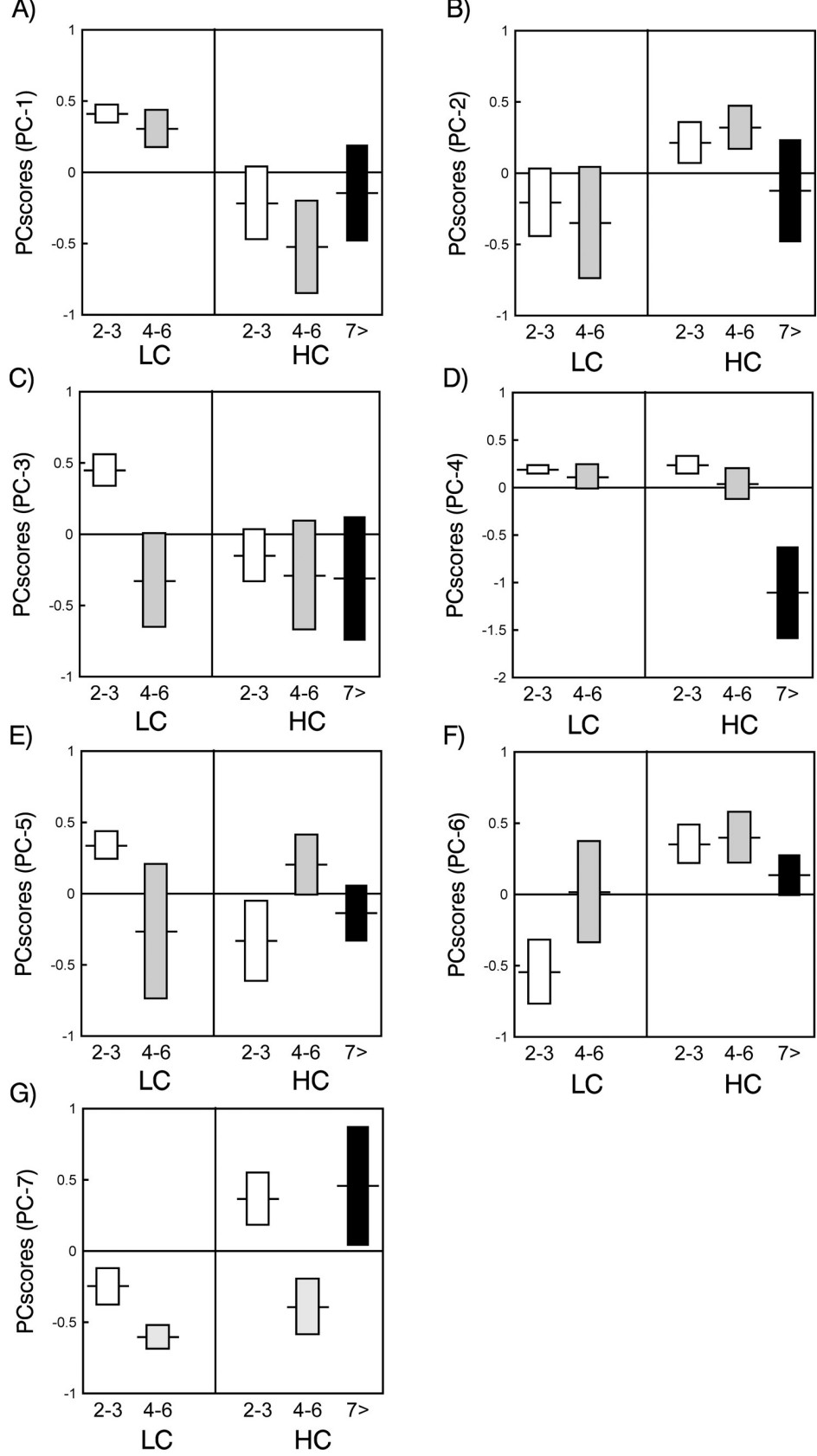

**Fig 1. Factor scores (mean ± SE) from a PCA (see Table 2) describing relative proportions of volatile compounds in the hair from dark ventral patch of Iberian red deer males (*Cervus elaphus hispanicus*) in relation to the intensity of male-male competition (LC: low; HC: high) and age categories (2–3 y: white, 4–6 y: grey; 7 and >7 y: black).**

compounds varied according to the age of the males, as it had already been found in a previous study [25], but, more interestingly, we found here that the chemical profiles also depended on the intensity of male-male competition in each population, independently of age variations. Aromatic compounds were less abundant in the older males, and in those individuals that experienced a lower intensity of intrasexual competition, while carboxylic acids were more abundant in males with a higher intensity of competition.

Differences in sexual gland chemical volatile composition due to age, sexual maturation and status rank have been largely demonstrated in mammals (e.g., *Elephas maximus* [40], *Ailuropoda melanoleuca* [41], *Mandrillus sphinx* [42–43], *Aotus spp*. [44], *Marmota marmota* [45]), including red deer, *Cervus elaphus hispanicus* [25]. However, the main aim of this study was to examine the differences in Iberian male red deer chemical profiles under two different situations referring to the intensity of male-male competition. Resource allocation in sex traits may vary according to environmental and individual conditions [46–47]. Here, we expected that the investment in chemical compounds that impregnate dark ventral patch hair may vary due to the age and status rank of the individuals, and also depending on the intra-sexual competition context in the population.

We found lower proportions of aromatic compounds and higher proportions of carboxylic acids in the hair of the ventral patch of males in a high competition situation than in males from low competition populations. Aromatic compounds with aromatic rings are very stable and are commonly found being part of range marks, signalling territoriality and social rank [48]. In elephant urine, aromatic compounds appear in higher proportions in urine from males in musth than in urine from non-musth males or from females [49]. Thus, our results are somehow contrary to the expectations by showing lower proportions of aromatic compounds in older adult males and in males from high-competition populations. However, among the aromatic compounds included in our analysis, the only one that significantly showed reduced proportions in adult males and in high-competition situations was *m*-cresol. This compound was abundant in elephants in musth [49], although it was not included among the aromatic ones reviewed by Apps et al. [48] for terrestrial mammals. Consistent with our results in this study, *m*-cresol was already found to be in lower proportion in the dark ventral patch of red deer in adult compared to young males [7].

On the other hand, fatty acids play an important role in chemical communication in several vertebrate species [50–52]. Particularly, oleic acid is often found in gland secretion of many species. For example, oleic acid is considering the only-one volatile compound involved in communication in water buffalos (*Bubalus bubalis*) [53], where it is present in vaginal mucus of oestrus females and in male saliva after licking the females' genital region, acting as sexual stimulant during the reproductive period. Oleic acid is the main compound used to store fat reserves (= energy) in the body of vertebrates and its allocation to gland secretions may be costly and dependent on the body condition of the male that produces the chemical signal [54]. Males in a high intra-sexual competition situation have higher proportions of oleic acid in their bellies compared to LC individuals, suggesting that HC males may invest more in signalling the amount of this compound, as it might reflect the body condition and the amount of body fat reserves. Anal gland secretion of dominant alpine marmot males also presents high concentrations of carboxylic acids, among other compounds [45]. According to our assumption, this result suggests a high investment in status rank marks in the intense intrasexual

competition context. Chemosignals can warn of the physical condition of males to individuals of the same sex, as reported for the North American elk (*Cervus canadensis*) [55]. This may be important to increase their success in competition against other males, which might also be reinforced if female preferences are concordant with the same male traits [56–57], but see also [58].

We found that LC males have higher proportions of androstane-3,17-dione (= androstane-dione) in their belly, while 3-hydroxy-5-androstan-17-one (= epiandrosterone) appears in higher proportions in HC males. Androstanedione is a steroid metabolite and a precursor of both testosterone and estrone, so it might have the function of signalling the testosterone level or the reproductive status of the individual, acting as a powerful signal [25]. Epiandrosterone has been identified as the major testosterone metabolite in a variety of male mammals (*Sus scrofa* [59], *Macaca fascicularis* [60], *Cebus capucinus* [61], *Crocuta crocuta* [62], *Acinonyx jubatus* [63]). It has been used as a gonadal activity indicator and the measurement of epiandrosterone metabolite reliably indicates the testicular endocrinal activity in ungulates (*Giraffa camelopardalis* [64], chamois, *Rupicapra rupicapra rupicapra* [65], *Cervus elaphus* [66]). Because of the low intrasexual competition experienced by LC males, these individuals might have more chances to mate than HC males. Under this situation, it might be more important to signal with androstanedione their reproductive status instead of their dominance rank. In contrast, we speculate that the higher proportions of epiandrosterone in HC males, related to gonadal activity, could reveal their quality and testicular activity in order to increase their success under an intense sexual selection situation. Additional research is needed to determine if fecal epiandrosterone metabolite levels are higher in HC than in LC males, and, in this case, how different is the gonadal activity in males from both scenarios.

According to previous studies [25], we observed notable differences in compounds found in the dark ventral patch hair between age categories. Benzoic acid was found in a greater proportion in individuals above 7 years, while cholesterol appeared in greater proportions in deer from 2 to 6 years and decreased in males older than 7. The *m*-cresol appeared more abundantly in young individuals, between 2 and 3 years, decreasing in older individuals. However, interestingly, this reduction was delayed in LC individuals with respect to HC males. Due to the bias towards females in the LC populations, male-male competition is not intense, which might render not so relevant to produce age variations in the chemical profiles, maintaining them with the characteristic of young individuals. Conversely, in HC populations, even young individuals are affected by male-male competition, which might result in young males having chemical profiles more similar to those of mature males.

These age differences in chemical profiles have also been evidenced in other species. Males of white-tailed deer secrete 46 compounds by the interdigital gland. Of these, 11 appear in greater concentration in older or dominant males than in young males or subordinates [10]. Moreover, it was demonstrated that in male whitetails, the urinary volatile compounds are different between subordinate and dominant males, probably revealing their hormonal status [13].

These differences in the proportions of chemical compounds found in the hair from the dark ventral patch between HC males and LC males support the idea that the intensity of male-male competition has an effect in the characteristics of secondary sexual traits. It is likely that the intensity of male-male competition may also have an effect on the kind of compounds (e.g., more or less volatile) that an individual allocates to the gland secretions and, consequently, that appear in the dark ventral patch, in order to increase the intensity and the duration of the chemical signal. According to this hypothesis, future studies should examine the costs (e.g., in terms of physiological stress) associated to the allocation of different chemical compounds to the hair from the ventral patch, and how these costs are related to the levels of

male-male competition. It would be also interesting to examine whether and how these different signalling strategies can affect current reproductive success but also compromise future reproductive success and even life span.

We conclude that Iberian male red deer modulate the expression of volatile compounds that are present in the dark ventral patch, according not only to its own condition, but also to the age and dominance status of the other males they live with. Males in a high male-male competition situation invest in volatile compounds that can reveal their age, dominance status and that, in addition, enhance this signal. On the contrary, individuals living under low intrasexual competition conditions have chemical profiles more characteristic of young individuals. This study throws light to additional behavioural research in order to show which volatile compounds present in hair from dark ventral patch are communicating significant information, relevant in sexual selection.

## Acknowledgments

We thank two anonymous reviewers for their helpful comments and people from the Biology and Ethology group (University of Extremadura at Caceres) and from the Wildlife Research Unit (UIRCP, University of Cordoba) for their help in field work and sample collection. Financial suport came from projects CGL2013-48122-P and CGL2016-77052-P to JC.

## Author Contributions

**Conceptualization:** Eva de la Peña.

**Data curation:** Eva de la Peña, José Martín.

**Formal analysis:** Eva de la Peña, José Martín.

**Funding acquisition:** Juan Carranza.

**Investigation:** Eva de la Peña, José Martín, Juan Carranza.

**Methodology:** Eva de la Peña.

**Resources:** Eva de la Peña.

**Software:** Eva de la Peña, José Martín.

**Supervision:** José Martín.

**Validation:** José Martín.

**Writing – original draft:** Eva de la Peña.

**Writing – review & editing:** José Martín, Juan Carranza.

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
