## [Decision Letter · Decision Letter 0]

18 Jul 2019

PONE-D-19-16624

The intensity of male-male competition may affect chemical scent constituents in the dark ventral patch of Iberian male red deer

PLOS ONE

Dear Dr De la Pena

Thank you for submitting your manuscript to PLOS ONE. After careful consideration, we feel that it has merit but does not fully meet PLOS ONE’s publication criteria as it currently stands. Therefore, we invite you to submit a revised version of the manuscript that addresses the points raised during the review process. most of which are minor suggestions aiming to improve the general readability of the paper.

We would appreciate receiving your revised manuscript by Sep 01 2019 11:59PM. To enhance the reproducibility of your results, we recommend that if applicable you deposit your laboratory protocols in protocols.io, where a protocol can be assigned its own identifier (DOI) such that it can be cited independently in the future. For instructions see: http://journals.plos.org/plosone/s/submission-guidelines#loc-laboratory-protocols

We look forward to receiving your revised manuscript.

Kind regards,

Marco Apollonio

Academic Editor

PLOS ONE

Journal Requirements:

1. We note that you have indicated that data from this study are available upon request. PLOS only allows data to be available upon request if there are legal or ethical restrictions on sharing data publicly. For more information on unacceptable data access restrictions, please see http://journals.plos.org/plosone/s/data-availability#loc-unacceptable-data-access-restrictions.

2. Please include your tables as part of your main manuscript and remove the individual files. Please note that supplementary tables (should remain/ be uploaded) as separate "supporting information" files

Reviewers' comments:

Reviewer's Responses to Questions

**Comments to the Author**

1. Is the manuscript technically sound, and do the data support the conclusions?

Reviewer #1: Yes

Reviewer #2: Yes

2. Has the statistical analysis been performed appropriately and rigorously? 

Reviewer #1: Yes

Reviewer #2: Yes

3. Have the authors made all data underlying the findings in their manuscript fully available?

Reviewer #1: Yes

Reviewer #2: Yes

4. Is the manuscript presented in an intelligible fashion and written in standard English?

Reviewer #1: Yes

Reviewer #2: Yes

5. Review Comments to the Author

Reviewer #1: Dear authors,

I have great interest read you paper.

1. Introduction. Very well referred and up to standards. Given the focus and hunting competition the introduction is weak with respect to that. Please frame the competition better to enlighten the reader

2. Study area etc. Please add when in relation to the rut the bulls were shot. Could that - time - explain some of the unexplained variation and that older males had lower concentrations of some compounds? What if older males are active earlier during the rut, but shot at the same time or later?

3.Results ok

4. Discussion. OK, but what if older males are active earlier in the rut irrespective of level of competition?

All the best and good luck with your research.

Reviewer #2: Authors studied differences in the presence and concentrations of several chemical scent constituents in the dark ventral patch of Iberian red deer males. They analysed hairs of 84 males originated from two groups of populations related to the harvesting pressure which affected demographic structure of both groups and consequently also caused differences in intraspecific competition, i.e. forming high vs. low male-male sexual competition situations. In these two groups, they studied differences in chemical scent constituents in dark ventral patches of males, which are formed during the rut period and contain volatile compounds that are used in intraspecific communication.

The influences of the age and the dominance of red deer males (and males of some other species) have already been studies several times as is recognized also by the authors of the manuscript; however, the influence of the intensity of the intrasexual competition on the composition of the scent compounds (and hence on the intraspecific communication) has not been studied so far. By finding two different groups of red deer considering their demographic structures and former knowledge on the reproductive efforts of red deer stags in both groups authors managed to find relevant situation in situ allowing also studying of the influence of the male-male sexual competition on the extraction of volatile chemical compounds. They found significant differences in concentrations/proportions of several compounds, and managed to provide logical explanations to this variation in relation to differences in intersexual competition among stags of both groups.

I strongly believe that the paper is worth to be published in the PLOS ONE after the minor revision, i.e. after correction of some mistakes that are stated below:

Title: “Iberian male red deer” should be changed either to “Iberian red deer males” or to “male Iberian red deer”.

Line 101: please, delete double space after “[25]”

Line 138: “estates” instead of “states”

Line 139: “in study areas” instead of “in these populations”

Line 155: “vs.” instead of “vs”

Line 157: add “,” after “competition”

Line 161: delete “recently”

Line 162: “estates” instead of “states”

Line 170: “or” instead of “of”

Line 210: delete “,” after “since”

Line 228: “0.5%” instead of “0.5 %”

Line 244: add “,” after “(5.22%)”

Line 275: please, use proper symbols “±”, and use “%” without the space after the number

Line 279: please, use proper symbols “±”, and use “%” without the space after the number

Line 288: “we compared” instead of “when comparing”

Line 313: delete “.” after “above”

Line 317: add space after “vs.”

Line 319: add “,” after “P = 0.0001)”

Line 320: add “, respectively” before “(Fig. 1)”

Line 402: please, write “Camelopardalis” in lower case

Line 426: delete “, Odocoielus virginatus,” (already used in the line 60, also misspelling)

Line 445: delete “(Cervus elaphus hispanicus)” (already used)

Line 452: “behavioural” instead of “behavioral”

References should be strictly edited according to the PLOS ONE standard; some mistakes are:

L467: “-“ at the end should be “.”;

L478: double “.”

L479: “:” should not be bolded

L481: “-“ instead of “–“

L483: “-“ instead of “–“

L491: double space before title

L511: “8” should not be bolded, webpage is mentioned only here

L527: double space before “79”

L528: change “,” after “Carranza” into the empty space

L529: “-“ instead of “–“

L541: “-“ instead of “–“

L580: “-“ instead of “–“

L595: “-“ instead of “–“

L597: “2006” should not be in italic; “-“ instead of “–“

L600: “-“ instead of “–“

L611: without “(4)”

L633: “2016” should not be bolded

L642: “.” should not be bolded

Figure legend:

Line 650: “4-6” instead of “4.6”

Line 651: “y:” instead of “y :”

Title of the Table 2: “0.5%” instead of “0.5 %”

6. PLOS authors have the option to publish the peer review history of their article (what does this mean?). If published, this will include your full peer review and any attached files.

Reviewer #1: No

Reviewer #2: No

---

## [Author Response · Author response to Decision Letter 0]

31 Jul 2019

Answers to associated-editor comments

1. We note that you have indicated that data from this study are available upon request. PLOS only allows data to be available upon request if there are legal or ethical restrictions on sharing data publicly. For more information on unacceptable data access restrictions, please see http://journals.plos.org/plosone/s/data-availability#loc-unacceptable-data-access-restrictions.

>>> Thank you for the information. We have no ethical or legal restriction, so we upload the data set to the FLGSHARE repository. We address this information in the cover letter.

>>> We have added our tables in the manuscript.

Answers to reviewer #1 comments

Reviewer #1: 

Dear authors,

I have great interest read your paper.

1. Introduction. Very well referred and up to standards. Given the focus and hunting competition the introduction is weak with respect to that. Please frame the competition better to enlighten the reader.

>>> We have added a paragraph about how the mate competition mediate the trait expression and hormone levels in male Iberian red deer. Also, we comment its implications in costs in terms of stress and immune function, taking into account the mating effort (e.g. testosterone levels) and the investment in sexual traits, such as antlers length and the dark ventral patch size.

2. Study area etc. Please add when in relation to the rut the bulls were shot. Could that - time - explain some of the unexplained variation and that older males had lower concentrations of some compounds? What if older males are active earlier during the rut but shot at the same time or later?

>>> Thank you for your comments. We added the specific date that the samples were collected in each population in the “Study area” epigraph. In all cases, these dates are not far from the rutting season (the most faraway case it is a month from the mating season), so we did not expect that this could have an influence on compounds concentrations between individuals of different age categories. It is worth to highlight that the date of sampling the study populations were not chosen using any criteria, but they were randomly sampled just depending on the hunting activity.

However, we agree with you that older males are active earlier in the rut than subadults or young males. But older males do not cease to be active earlier than these other younger individuals. Hence, we do not expect differences in compounds between age categories due to the same passage of time.

3. Results. OK.

4. Discussion. OK, but what if older males are active earlier in the rut irrespective of level of competition?

>>> Despite it could be true that older males were active in the rut a little bit earlier than subadult and young males, we would expect to find similar temporal pattern of the loss or volatilization of compounds that impregnate the fur in both competition scenarios. However, we found differences in compound profiles between intrasexual competition level at the same category age. 

All the best and good luck with your research.

>>> Thank you very much.

Answers to reviewer #2 comments

Reviewer #2: 

Authors studied differences in the presence and concentrations of several chemical scent constituents in the dark ventral patch of Iberian red deer males. They analysed hairs of 84 males originated from two groups of populations related to the harvesting pressure which affected demographic structure of both groups and consequently also caused differences in intraspecific competition, i.e. forming high vs. low male-male sexual competition situations. In these two groups, they studied differences in chemical scent constituents in dark ventral patches of males, which are formed during the rut period and contain volatile compounds that are used in intraspecific communication.

The influences of the age and the dominance of red deer males (and males of some other species) have already been studies several times as is recognized also by the authors of the manuscript; however, the influence of the intensity of the intrasexual competition on the composition of the scent compounds (and hence on the intraspecific communication) has not been studied so far. By finding two different groups of red deer considering their demographic structures and former knowledge on the reproductive efforts of red deer stags in both groups authors managed to find relevant situation in situ allowing also studying of the influence of the male-male sexual competition on the extraction of volatile chemical compounds. They found significant differences in concentrations/proportions of several compounds and managed to provide logical explanations to this variation in relation to differences in intersexual competition among stags of both groups.

I strongly believe that the paper is worth to be published in the PLOS ONE after the minor revision, i.e. after correction of some mistakes that are stated below:

Title: “Iberian male red deer” should be changed either to “Iberian red deer males” or to “male Iberian red deer”. >>> OK, changed

Line 101: please, delete double space after “[25]” >>> OK, changed

Line 138: “estates” instead of “states” >>> OK, changed

Line 139: “in study areas” instead of “in these populations” >>> OK, changed

Line 155: “vs.” instead of “vs” >>> OK, changed

Line 157: add “,” after “competition” >>> OK, changed

Line 161: delete “recently” >>> OK, changed

Line 162: “estates” instead of “states” >>> OK, changed

Line 170: “or” instead of “of” >>> OK, changed

Line 210: delete “,” after “since” >>> OK, changed

Line 228: “0.5%” instead of “0.5 %” >>> OK, changed

Line 244: add “,” after “(5.22%)” >>> OK, changed

Line 275: please, use proper symbols “±”, and use “%” without the space after the number >>> OK, changed

Line 279: please, use proper symbols “±”, and use “%” without the space after the number >>> OK, changed

Line 288: “we compared” instead of “when comparing” >>> OK, changed

Line 313: delete “.” after “above” >>> OK, changed

Line 317: add space after “vs.” >>> OK, changed

Line 319: add “,” after “P = 0.0001)” >>> OK, changed

Line 320: add “, respectively” before “(Fig. 1)” >>> OK, changed

Line 402: please, write “Camelopardalis” in lower case >>> OK, changed

Line 426: delete “, Odocoielus virginatus,” (already used in the line 60, also misspelling) >>> OK, changed

Line 445: delete “(Cervus elaphus hispanicus)” (already used) >>> OK, changed

Line 452: “behavioural” instead of “behavioral” >>> OK, changed

References should be strictly edited according to the PLOS ONE standard; some mistakes are:

L467: “-“ at the end should be “.”; >>> OK, changed

L478: double “.” >>> OK, changed

L479: “:” should not be bolded >>> OK, changed

L481: “-“ instead of “–“ >>> OK, changed

L483: “-“ instead of “–“ >>> OK, changed

L491: double space before title >>> OK, changed

L511: “8” should not be bolded, webpage is mentioned only here >>> OK, changed

L527: double space before “79” >>> OK, changed

L528: change “,” after “Carranza” into the empty space >>> OK, changed

L529: “-“ instead of “–“ >>> OK, changed

L541: “-“ instead of “–“ >>> OK, changed

L580: “-“ instead of “–“ >>> OK, changed

L595: “-“ instead of “–“ >>> OK, changed

L597: “2006” should not be in italic; “-“ instead of “–“ >>> OK, changed

L600: “-“ instead of “–“ >>> OK, changed

L611: without “(4)” >>> OK, changed

L633: “2016” should not be bolded >>> OK, changed

L642: “.” should not be bolded >>> OK, changed

Figure legend:

Line 650: “4-6” instead of “4.6” >>> OK, changed

Line 651: “y:” instead of “y :” >>> OK, changed

Title of the Table 2: “0.5%” instead of “0.5 %” >>> OK, changed

>>> Thanks for your comments and corrections, we are very grateful for it.

---

## [Editor Report · Decision Letter 1]

20 Aug 2019

The intensity of male-male competition may affect chemical scent constituents in the dark ventral patch of male Iberian red deer

PONE-D-19-16624R1

Dear Dr. De la Pena,

We are pleased to inform you that your manuscript has been judged scientifically suitable for publication and will be formally accepted for publication once it complies with all outstanding technical requirements.

With kind regards,

Marco Apollonio

Academic Editor

PLOS ONE
---

## [Editor Report · Acceptance letter]

23 Aug 2019

PONE-D-19-16624R1 

The intensity of male-male competition may affect chemical scent constituents in the dark ventral patch of male Iberian red deer 

Dear Dr. De la Peña:

I am pleased to inform you that your manuscript has been deemed suitable for publication in PLOS ONE. Congratulations! Your manuscript is now with our production department. 

With kind regards,

on behalf of

Prof. Marco Apollonio 

Academic Editor

PLOS ONE